Acoustic and visual monitoring of Peale’s dolphins (Lagenorhynchus australis) in the Magellan Strait

Malige Franck 1 2
Patris Julie 1 3 julie.patris@univ-amu.fr
http://orcid.org/0000-0002-6356-3770 Caceres Benjamin 4
Poblete Jonathan 4
Glotin Hervé 1 2 5
Saravia Monserrat 4
Alarcón-Vera Constanza 4
http://orcid.org/0000-0003-1814-7679 Barchasz Valentin 5
http://orcid.org/0000-0002-2099-1150 Gies Valentin 5
Marzetti Sebastian 5
Fuentes-Riquelme Marjorie 6 7
Filún Diego 8 9
1 CIAN, International Center of Artificial Intelligence in Natural Acoustics , Toulon , France
2 Toulon University, DYNI Team, LIS Laboratory, CNRS UMR 7020 , Toulon , France
3 Aix-Marseille University, DYNI Team, LIS Laboratory, UMR 7020 , Marseilles , France
4 Asociación de Investigadores del Museo de Historia Natural Río Seco , Punta Arenas, region de Magallanes , Chile
5 SMIoT, Toulon University , Toulon , France
6 NGO Centro de Estudios Para la Conservación de Ecosistemas Marinos , Santiago , Chile
7 NGO Yaqu Pacha , Nuremberg , Germany
8 Alfred Wegener Institute , Bremerhaven , Germany
9 Centro Ideal, Universidad austral de Chile , Punta Arenas , Chile
Ereskovsky Alexander
Electronic publication date: 2025 Jun 26
Publication date: 2025
Volume: 13
Electronic Location ID: e19234
Received 2024 Dec 12; Accepted 2025 Mar 10
Copyright: © 2025 Malige et al.
Copyright year: 2025
Copyright holder: Malige et al.
License: This is an open access article distributed under the terms of the Creative Commons Attribution License, which permits unrestricted use, distribution, reproduction and adaptation in any medium and for any purpose provided that it is properly attributed. For attribution, the original author(s), title, publication source (PeerJ) and either DOI or URL of the article must be cited.
License URL: https://creativecommons.org/licenses/by/4.0/

Keywords: Bioacoustics, Peale’s dolphin, High frequency, Remote sensing

Funding: University of Toulon, France NGO Yaqupacha, Germany The Rio Seco Natural history Museum, Region de Magallanes, Chile 55c05f-1 Materials were financed by the University of Toulon, France and the NGO Yaqupacha, Germany (loan of material). The Rio Seco Natural History Museum, Region de Magallanes, Chile, organized and financed the field work and the hosting of the researchers with the Rufford project number 55c05f-1, “Seasonality, moving patterns and identification of the Endangered Sei Whale (Balaenoptera borealis) Population in the Strait of Magellan, Chile”. The funders had no role in study design, data collection and analysis, decision to publish, or preparation of the manuscript.

==============================
Peale’s dolphins are small odontocetes living in the south Cone of America. As a coastal species, they could be threatened by the growing use of industrial and comercial harbors in the Patagonia region. To the date, their behavior, use of habitat and acoustic productions are still very little known. We report the coupling of visual and acoustic monitoring of a population of Peale’s dolphins in the Magellan Strait. A comparison of visual monitoring and data from several acoustic devices (three C-POD click loggers and HighBlue, a full wave recording device) is presented, assessing the feasibility of long term monitoring through acoustic recording. Finally, acoustic productions of this group of Peale’s dolphins are compared to other groups of Peale’s dolphins and other narrow band high frequency (NBHF) odontocetes of the region.

Introduction

Chilean Patagonia is a vast and ecologically diverse region, home to a wide variety of marine species, including several cetaceans (Gibbons, Gazitua & Venegas, 2000; Aguayo-Lobo, Acevedo & Olave, 2007; Capella & Gibbons, 2020; Viddi, Bedriñana-Romano & Hucke-Gaete, 2023). Its remote, rugged coastline and complex marine environment have made it challenging to study and monitor marine wildlife. In recent years, anthropogenic expansion (including increased shipping, aquaculture, and coastal development) has raised concerns about the potential impacts on local ecosystems and species (Buschmann et al., 2009; Gonzalez et al., 2010; Bedriñana-Romano et al., 2021, 2023). Monitoring these changes and their effects on marine fauna, particularly cetaceans, is critical for developing effective conservation and management strategies.

Peale’s dolphins (Lagenorhynchus australis) is a species of small cetacean native to the cold temperate waters of southern South America, particularly the coastal areas of the Magellan Strait and the Patagonian region (Viddi & Lescrauwaet, 2005; Heinrich, 2006; Viddi et al., 2010; Dellabianca et al., 2016; Heinrich et al., 2019). Despite their ecological importance and unique adaptations to this remote environment, the study of Peale’s dolphins has been hindered by the logistical challenges posed by the region’s rugged terrain and harsh weather conditions. Traditional survey methods, such as visual observations or boat-based studies, are often limited in effectiveness due to these factors, leaving significant gaps in our understanding of this species’ behavior, distribution, and acoustic communication (Patris et al., 2023).

Passive acoustic monitoring (PAM) presents a valuable, non-invasive alternative to overcome the limitations of traditional survey methods. By deploying autonomous acoustic devices at remote and hard-to-reach sites, such as Chilean Patagonia, researchers can continuously monitor the vocalizations of marine species over extended periods (Gibb et al., 2018). This approach offers critical insights into species acoustic presence, movement patterns, habitat use, and the potential effects of human activities, all without requiring constant human presence. This is especially true for odontocetes species, which usually emit clicks used for ecolocation as well as communication sounds. PAM is particularly effective in regions where visual surveys are impractical due to logistical constraints, enabling medium or long-term monitoring in challenging environments (Amundin et al., 2022; Filun & Opzeeland, 2023; Patris et al., 2023). However, the long term monitoring of delphinids by PAM techniques is rendered difficult by the frequency characteristics of their acoustic emissions.

The Peale’s dolphin is characterized as a narrow band, high frequency (NBHF) species (Rojas-Mena, 2009; Kyhn et al., 2010; Martin et al., 2024). The classic sonar-type emission, or “click”, of NBHF species has a dominant frequency of about 130 kHz and a quality factor ( Qrms, see Supplemental Materials for definition) larger than 5 (Kamminga, Cohen-Stuart & Silber, 1996). There are sixteen species in the world known to emit NBHF clicks, which are thought to be a convergent evolution of small odontocetes (as diverse as porpoises and dolphins) to escape the notice of predators (Morisaka & Connor, 2007). Until few years ago, such species were supposed to emit only high frequency clicks but, recently, a few studies (Martin et al., 2018, 2021; Nielsen et al., 2024; Martin et al., 2024) have shown that some of these species can emit clicks at lower frequency (“relaxing the acoustic crypsis”) and also whistles, though maybe not frequently. NBHF species are distributed all over the world, but are especially important in Southern Patagonia. Indeed, it is one of the few places where several NBHF species are sympatric and Peale’s dolphins often shares the same environment with other species emitting similar sounds: Chilean dolphins (Cephalorhynchus eutropia), Commerson’s dolphins (Cephalorhynchus commersonii) and Burmeister porpoise (Phocoena spinipinnis).

The very high frequency of NBHF clicks makes it difficult for standard instruments to record the full waveform, and specialized instruments with a sampling rate higher than 300 kHz are very demanding in energy and storage capacity. Thus the autonomy of such high frequency instruments is usually limited to a few weeks, as a maximum (Gillespie et al., 2020; Patris et al., 2023). As an alternative, C-PODs (The Cetacean Porpoise Detector, Tregenza (2014), now replaced by the new generation F-POD) are affordable and easy-to-use devices that allow continuous monitoring over periods ranging from 3 to 6 months. Though the device does not capture the full wave sound, it does register the date and some parameters of individual echolocation clicks within the 20 to 160 kHz range of most odontocetes, and has become widely used in research on dolphins’ and porpoises’ behavior and ecology across the globe (Robbins et al., 2015; Sarnocinska et al., 2016; Jacobson et al., 2017; Amundin et al., 2022; Paitach et al., 2023). The aim of our research is twofold. On the one hand, we assess methods of studying these species by comparing the effectiveness of C-PODs and high-frequency hydrophones, while also evaluating their performance relative to visual sighting methods. Our research focuses on the potential of passive acoustics for studying Peale’s dolphins in remote locations, particularly in the Magellan Strait and Patagonian regions. On the other hand, we review the current knowledge of the Patagonian NBHF species’ vocal repertoire and discuss the broader implications of acoustic data for the conservation and management of this species in a rapidly changing environment.

Materials and Methods

Visual and acoustic monitoring

In December 2021, during 4 days from the 7th, 18 h local time (UTC-3 h) to the 11th, 11 h local time, we implemented a visual (18 h) and acoustic (90 h) survey of small odontocetes. The location is a little bay at the north of San Isidro lighthouse, in the Magellan Strait, 60 kms south of Punta Arenas, Chile (see Fig. 1, top left). The bay opens to the North-East, protected from the dominant wind, coming from the west, and hosts various kelp beds, known to be a typical habitat for the Peale’s dolphin (Viddi & Lescrauwaet, 2005). It is also a very remote area, accessible only with a small boat or on foot by a little coastal trail. The observing team (F. Malige, J. Patris, B. Caceres, J. Poblete and M. Saravía) were accommodated in a private house with a view over the bay.

Figure 1 Top left: Localization of our study and previous acoustic studies on Peale’s dolphins (Lagenorhynchus australis). Top right: Set up of San Isidro acoustic experiment (three C-PODs and HighBlue device). The two mooring lines are independent: the buoy marks the place for visual follow-up, but in case it should be removed (by people or bad meteorological conditions) the instruments will not be affected. Bottom left: “HighBlue device”, in Puyuhuapi Fjord in may 2021, (credit: Cristian Maldonado), Patris et al. (2023). Bottom right: Peale’s dolphin, San Isidro on December, 10th 2021. The buoy of the experiment is visible in the foreground (Photo Jonathan Poblete).

For the visual survey, we monitored the bay during 3 days (8th, 9th and 10th of December), by groups of two observers equipped with binoculars, 6 h each day (9h30–12h30/16h30–19h30, local time). We systematically noted each visual event (that is the following of the same groups of animals), its beginning and ending hour, the species, the number of individuals estimated (groups size), their behavior (slow travel, fast travel, foraging, social interactions) and any relevant additional comment. As the group of researchers was living in the house with a view over the bay, we also noted all the “opportunistic events”, i.e., visualization of a group of odontocetes in the bay outside the hours of systematic visual survey. A drone was available but could not be used because of complicated meteorological conditions (high wind). During the experiment, we define “daylight” as the period between 5h10 local time (8h10 UTC) and 22 h local time (1h00 UTC) (source: https://www.imcce.fr/).

For the acoustic survey, three click detectors (C-PODs devices, Tregenza (2014)) and a High Blue recording system (Barchasz et al., 2020) were installed in a zone with approximately 10 m of water depth and situated at 100 m from the shore ( 53o46′53,46′′ South/ 70o58′18,9′′ West). They were installed on Dec, 7th at 18 h local time and retrieved on Dec, 11th at 11 h local time. The three C-POD, property of NGO Yaqupacha who lent them, were installed together in order to compare their sensitivities, and compare them to a full wave recording device, in the mark of a long-term monitoring of the coast North of Punta Arenas (B. Caceres, ongoing work). The C-PODs are positively buoyant, and were installed in a mooring line at respectively 2, 4 and 6 m from the sea floor (see Fig. 1, top right).

The recording instrument ‘HighBlue’ (https://smiot.univ-tln.fr/index.php/highblue/), installed on the sea floor, was built by the Scientific Microsystems for the Internet of Things (SMIoT) technical team from Toulon University. It contains a digital data acquisition system, Qualilife HighBlue (QHB), that allows receiving several signals in parallel with very low consumption (2 to 4 W). The sampling rate was chosen at its maximum (512 kHz) and the dynamics at 16 bits to have more autonomy in storage. Data storage is done on a 512 Gb μSD card. The system is powered by 21 D2-type batteries (24 kWh capacity). The hydrophone used is a C57 hydrophone from Cetacean Research (http://cetaceanresearch.com/), with a cylindrical transducer (omnidirectional under 10 kHz, sensitive in the median plane for higher frequencies). Its frequency sensitivity is linear within −12 dB from 8 Hz to 100 kHz (manufacturer’s documentation) and up to more than 150 kHz according to our subsequent calibrations. Although two hydrophones can be used, only one was installed for lower energy and memory consumption (see Fig. 1, bottom left). For technical reasons, inherant to this version of the QHB acquisition card, we chose a recording duty cycle of 9 min ON, 1 min OFF. The electronics is contained in a transparent Plexiglas tube, resistant to pressure up to 100 meters, equipped with two custom built legs and two supports for the hydrophones. With the chosen parameters, the device has a battery life of more than 7 days and 5.5 days for memory, sufficiently for our experiment: recording for about 4 days (from December 7 to 11, 2021) without needing to replace the memory card or batteries.

Detections

We programmed a click detector in OCTAVE (Eaton, Bateman & Hauberg, 2009) to find narrow band high frequency (NBHF) clicks in the HighBlue data set, the same used in Puyuhuapi canal and presented in Patris et al. (2023) (see Supplemental Materials for details). We estimate the false negative rate around 60% and the true positive rate around 95%. Thus, although a large number of real clicks are discarded, the detections corresponds to real NBHF clicks at 95%. We noticed that this number of false positives usually corresponds to around 10 false detections for each file, mainly due to noise. Thus, we considered a 9-min WAV file as containing clicks only if it contains more than 20 detections. In addition, we manually revised all the intervals of 20s centered in each detection in all the files with at least 20 detections. This revision was done using spectrogram in Audacity software (Hann window of 256 points, 50% overlap). In this step, we systematically noted the eventual presence of quick trains of clicks (defined as trains where the interval between to clicks is less than 5 ms). Up to very recently, NBHF dolphins were not considered to emit other sound than NBHF clicks. However, Martin et al. (2024) reported the detection of whistles emitted by Peale’s dolphins in Falkland Islands. Thus, we also checked manually for the presence of whistles in the files containing clicks detections (spectrograms between 0 and 20 kHz, Hann window of 8,192 points, 50% overlap). Unfortunately, we were not able to revise manually all files, thus, if whistles have been emitted without clicks, we could not detect them.

A computation of the number of clicks by chunks of 10 min were achieved for the three C-PODs by mean of the software provided by Chelonia (Tregenza, 2014). In the Highblue files, the number of clicks per 10-min chunks are biased by our non continuous mode of recording (9-min ON, 1 min OFF). However, this bias is of little consequence given the high variability of the detection rate of all four instruments. We defined a “detection event” as a subset of consecutive 10 min-files where clicks were detected (for HighBlue device, we considered only files with more than 20 detections). We also compared the click rate (average number of clicks by hour) in each event in function of the group size, visually estimated. We then compared it with theoretical models of click production: a linear model where clicks are produced independently of other dolphins, a quadratic model where clicks are produced at a constant rate depending on the number of other dolphins and a mixed model.

Clicks parameters

For each detected click, we measured, as presented in Patris et al. (2023), the classical acoustic parameters of the detected signals: peak frequency, centroid frequency, rms bandwidth, −3 dB bandwidth, −10 dB bandwidth, rms duration, −10 dB duration, −20 dB duration and inter-click interval (ICI) in order to compare these values with values found in previous studies. The definitions used to compute these parameters are given in the Supplemental Materials. For further analysis, we also selected two subsets of the detected clicks: the subset of clicks for which the species is visually confirmed and the subset of supposedly “on-axis” clicks, considering the following definitions. We consider acoustic data to be “visually confirmed” when a click detection event is within 10 min of a sighting for which the species is confirmed. We took into account all sightings from December 7th to 11th. That is, systematic visual monitoring or opportunistic sightings were used to confirm the species. Dolphin clicks are usually very directional. In this case, the parameters of the emissions can strongly depend on the direction of reception of the signal (Macaulay et al., 2020). To mitigate the effect of the emission lobe, some studies (for example Kyhn et al. (2010)) select only the loudest click of a train, assuming that they are the most “on axis” clicks possible. This hypothesis seems debatable to us, however, for comparison purposes, we automatically selected a subset of clicks in the following way: among the clicks selected by the detector, series of at least five clicks are identified, separated by at least 0.2 s from the next series. In each train or series, only the click with the largest amplitude is selected as being “on-axis”, following Kyhn et al. (2010), Martin et al. (2024). Finally all the results of parameters were systematically compared with acoustic studies of Peale’s dolphins (Rojas-Mena, 2009; Kyhn et al., 2010; Reyes Reyes et al., 2018; Martin et al., 2024) as well as with acoustic studies of other NBHF species in Patagonia (Reyes Reyes et al., 2015; Patris et al., 2023).

Results

Visual monitoring

During the visual monitoring (18 h out of 90 h of acoustic recording), the only odontocete species seen was Peale’s dolphin. There were two sightings of groups of Peale’s dolphins during the monitoring on December 8th (a day with a lot of wind and rough sea), five sightings on December 9th and seven sightings on December 10th. In addition to these sightings, a total of 11 opportunistic sightings of Peale’s dolphins were noted between December 7th at 20 h UTC and December 11th at 13 h UTC. No other species of odontocetes were seen during the 3 days of the experiment. Although Chilean dolphins (Cephalorhynchus eutropia) or Commerson dolphins (Cephalorhynchus commersonii) have been seen occasionally in the Strait of Magellan (Gibbons, Gazitua & Venegas, 2000; Viddi, Bedriñana-Romano & Hucke-Gaete, 2023), among the species emitting NBHF clicks, only Peale’s dolphins have been reported in this bay in the last years (B. Caceres, 2021, personal observations). Therefore, we are confident that all the NBHF clicks recorded are from Peale’s dolphins. Groups of one to five individuals were sighted sometimes including one or two juveniles (11% of the visual events). The most important behavior was slow travel (88% of the visual events), but foraging (3%) or fast moving and socializing behavior (9%) were also recorded, notably when two groups (of four and five dolphins) met near the buoy, one coming from the north and the other from the south on the 10th of December, 21h31 local time (see Fig. 1, bottom right). Opportunistically, Magellanic penguins (Spheniscus magellanicus), a sea otter (Lontra felina) and sea lions (Otaria flavescens) were also seen near the buoy and we sighted one Sei whale (Balaenoptera borealis) several kilometers from the coast (8th of December, 18h52 local time). During all the duration of the experiment no strong current was noticed in the bay, that could have changed the depth of the instruments.

Acoustic monitoring

The “HighBlue” device recorded 536 acoustic files of 553 Mb each from the moment of its installation until its recovery, amounting to around 90 h of recording (3 days and 17 h) and 296 Gb of full wave data. All three C-POD instruments also worked without any problem during the 90 h of the experiment.

Number of detections and events

The automatic detector found 14,451 clicks in the HighBlue data set (corresponding to around 13,700 true clicks, taking into account the true positive rate of 95%). It is the largest data set of Peale’s dolphins clicks obtained so far (Reyes Reyes et al., 2018; Martin et al., 2024). After visual examination, three consecutive very noisy 9-min WAV files, with less than 25 detections in each, containing only false positives, were removed from the analysis after visual examination of their spectrogram (files 20211208_204215UTC.wav, 20211208_205216UTC.wav, 20211208_210217UTC.wav). The C-POD no 2150 (placed higher in the column of water) detected around 8,000 NBHF clicks, the C-POD no 2154 (at the center) detected around 6,000, and the C-POD no 2155 at the bottom detected around 9,000. The number of detection events was 44 for the HighBlue device, 43 for the C-POD no2150, 40 for the C-POD no2154, and 41 for the C-POD no2155. The number of acoustic events by day is rather similar between instruments (around 11 events per day). The dates of events are consistent between the two types of instruments (data logger and full recording systems), as in Sarnocinska et al. (2016), Patris et al. (2023) (see Fig. 2). There was a total of 48 distinct acoustic events. Of these 48 events, 35 were detected on the four instruments (73%) and 42 on more than two instruments (88%).

Figure 2 Click detections for periods of 10 min between December 7 and 11, 2021.

On the abscissa, the recording hours from the beginning of the experiment (17h40 Local Hour/20h40 UTC). The graph above represents the detections of the HighBlue device, the next three respectively those of the CPODs 2150, 2154 and 2155. The graph below represents the number of individuals in the visual detections, the red areas showing moments of continuous effort. Outside of these periods, sightings are opportunistic. The nights are with a gray background. The average number of clicks by hour (during the day or during the night) is given at the right of the time series, for each device.

Characteristic of the emitted sound

Statistical distributions of the parameters measured for all the automatic detections of the HighBlue device are presented in Fig. 3.

Figure 3 Histograms of click parameters: peak frequency, centroid frequency, inter-click interval (ICI), ‘rms’ bandwidth, ‘−3 dB’ bandwidth, ‘−10 dB’ bandwidth, ‘rms’ duration, ‘−10 dB’ duration and ‘−20 dB’ duration, see Supplemental Materials for formal definitions.

No train of clicks with significant energy below 100 kHz were found neither in the automatic detections nor in the visual review. Some clicks of peak frequency between 150 and 180 kHz were found manually that were not detected by our automatic detector. These manually found clicks are not included in the statistics of Fig. 3. Twenty “quick trains” (ICI from 1.4 to 5 ms, frequency peak between 105 and 130 kHz, narrow band) were visually detected in 10 of the 65 HighBlue files containing more than twenty click detections. The clicks of these trains were in general of low SNR. These “quick trains” match the usual definition of “burst pulse” (Martin et al., 2018), that is an isolated train of clicks with inter-click interval lower than 10 ms, possibly ending with a growing ICI. An example of our “burst pulse” is given Fig. 4.

Figure 4 Representation of a quick train of clicks (“Burst pulse”), with its ICI (inter-clicks interval) variation.

The signal to noise ratio of the clicks is low compared to classical clicks. Spectrogram parameters: 1,024-pt FFT, Hanning window, 90% overlap, 512 kHz sampling rate; Octave.

From all automatically detected clicks, 4,860 (34% of the total) were classified as “with visual counterpart”, thus asserting the species. The parameters of this subset have similar statistical distributions as the total number of clicks. The average and standard deviation of the whole set of data as compared to the “species visually confirmed” subset are presented in Table 1, and the full presentation of the parameters’ statistics are presented in the Supplemental Materials. Among the detected clicks, 552 possibly “on-axis” clicks were selected (3.8% of the total). The “on-axis” clicks parameters also have a statistical distribution (given in the Supplemental Materials) and average and standard deviation (Table 1) similar to the total number of clicks.

Table 1 Parameters of the clicks recorded by the HighBlue device (average and standard deviation) compared to two subsets: the “species visually confirmed” and the “on-axis” clicks.

For the ICI, the average and standard deviation are calculated only for the values that are below 200 ms, according to the distribution in the Fig. 3, thus eliminating the ICI that correspond to two distinct trains.

Clicks	Total	Species visually confirmed	“On-axis”	
Number of clicks analyzed N	14,451	4,860	552	
Peak frequency	131 ± 11 kHz	132±11 kHz	132±14 kHz	
Centroid frequency	138 ± 9 kHz	139±9 kHz	138 ±10 kHz	
‘rms’ bandwidth	16 ± 5 kHz	17±4 kHz	15±5 kHz	
−3 dB bandwidth	5 ± 2 kHz	5± 2 kHz	5±2 kHz	
−10 dB bandwidth	12 ± 6 kHz	12±6 kHz	12±6 kHz	
‘rms’ duration	79 ± 46 μs	78 ± 41 μs	77 ± 45 μs	
−10 dB duration	74 ± 42 μs	67 ± 38 μs	77 ± 46 μs	
−20 dB duration	160 ± 83 μs	154 ± 79 μs	159 ± 80 μs	
Inter clicks interval (ICI)	69 ± 41 ms	73 ± 43 ms	72 ± 41 ms	

Daily pattern

The number of clicks emitted by hour is much higher during night than during day (see Fig. 2). Considering the four devices, the number of clicks by hour is 2.3 ± 0.2 higher by night than by day (mean and standard deviation). As for the number of events by hour, the ratio between night and day is 1.2 ± 0.2. Despite our having more or less the same number of acoustic events between night and day, the total number of clicks per hour is higher at night. Thus, acoustically, Peale’s dolphins in this spot seem more active at night. Results from a C-POD device installed to monitor Peale’s dolphins in another part of Magellan Strait (in Rio Seco, near Punta Arenas) shows the same tendency (B. Caceres, 2021, personal communication).

Other sounds

No whistle similar to those recorded by Martin et al. (2024) were detected by our manual revision of all the files with more than 20 click detection. However, two series of atypical low frequency vocalizations were visually detected (at 6′42″ and 7′25″ in the file 20211211_045835UTC.wav) along with typical NBHF clicks and a 115 kHz quick train or burst pulse (ICI around 2ms). These vocalizations are tonal (with a fundamental between 0.5 and 1 kHz and almost no modulation in frequency) and last between 0.5 and 1 s. They are rather similar to killer whales’ (Orcinus orca) or other larger odontocetes’ vocalizations, but no traditional broadband click were recorded. It is possible that these low frequency vocalizations were emitted by Peale’s dolphins (there was no visual confirmation, the recording being at night) even though they are quite different from the whistles recently mentioned for Peale’s dolphins (Martin et al., 2024).

Coupling visual and acoustic analysis

Although the number of detections varies between instruments, there is a good coherence in detections events for the four acoustic devices (see Fig. 2). Visual detections and acoustic detections are not identical (see Fig. 2). During the systematic visual survey, almost all acoustic events have a visual counterpart (12 of 13 i.e., 92%). Considering all the 25 visual events (including opportunistic sightings, outside of the 18 h of systematic visual survey), 19 have an acoustic counterpart (76%). In three of the six visual events without acoustic counterpart, the group of dolphins was detected far from the acoustics devices (more than 500 m). Thus, the range of efficiency of the acoustical setting should be carefully determined if trying to evaluate parameters of the population, such as density or habitat use.

Behavior

We measured the influence of group size on the number of clicks emitted each hour. The results are given in Fig. 5. Three models of click production rate can be contemplated. The first model considers that clicks are used individually for echolocation. In this case, the click rate should be proportional to the number of individuals. A second model considers the clicks have a function of communication between two individuals, thus, the click rate should be proportional to the number of pairs n(n+1)2where n is the group size. Thirdly, an hybrid of the two precedent models can also be tested. After fitting the three models to the data, we found that the linear model is less consistent with the data (the coefficient of determination is R2=0.88) than the two other models ( R2=0.98). Indeed, in Fig. 5, the number of clicks increases rapidly with the group size in a power law tendency. This result confirms that the role of clicks for the Peale’s dolphin is not only echolocation, and that it also plays a role in social intercourse.

Figure 5 Curve in red: Average and standard deviation of the number of clicks by hour recorded by the HighBlue device in function of the groupe size, estimated visually.

Curve in blue: Fit of a function α×n(n+1)2 where n is the group size. The value of α was fitted with the leasqr function in Octave and is α≃110 clicks/h/pairs.

We also checked the behavior associated with the “Burst Pulses” (or quick trains of clicks) found in the Highblue data. Of the 20 trains, six were during the night, seven during “slow displacement” and seven during “socializing” behavior. The number of animals during the moment of burst pulse emission was always greater than 3, amounting once to around nine individuals (encounter of two groups). Of the 21 events visually associated with displacement, two contained burst pulses, while of the two events associated with socializing, both contained burst pulses. Thus, our results are compatible with the traditional role attributed to burst pulses, which are supposed to have a social function (Martin et al., 2018).

Discussion

Acoustic and visual survey

Our results show that a lot of information can be obtained from coupling acoustics and visual monitoring on a medium-term basis. Compared to the effort, the volume of acoustical data is much larger than when the experiment is conducted from a boat, and with much less disturbance. Martin et al. (2024) underlined that the Peale’s dolphin behavior is much influenced by the presence of the boat, thus potentially biasing the acoustical and behavioral pattern. What’s more, night-time acoustical full-wave data is also available, which is not common in dolphin studies in the wild. Visual monitoring is necessary to assert the species, and try to describe behavior. However, though deciding the species from the shore is rather easy, qualifying the behavior is very tentative. That is why our description lacks details and is maintained to very simple categories. A very good tool to get more information about the behavioral state of the dolphins is a drone with a camera high enough to not disturb the animals (higher than 50 m for a quiet little drone seems a reasonable criteria, see Christiansen et al. (2016)). However, we were not able to follow up the animals with our drone, partly because of the quick displacement of the groups (for which a higher station of observation could have been useful) and partly because of the region’s violent winds.

Comparison of clicks parameters with other acoustic studies of Peale’s dolphins

The results of the measure of the parameters are presented in Table 2 along with the results of the only three other studies published on the acoustics of Peale’s dolphins (see Fig. 1): Rojas-Mena (2009) near the island of Chiloe (Chile); Kyhn et al. (2010) and Martin et al. (2024) near the Falkland Islands (United Kingdom). The results of an unpublished study by Marino et al., cited in Reyes Reyes et al. (2018), were also added.

Table 2 Parameters of the clicks recorded by the HighBlue device (average and standard deviation) compared to other studies on Peale’s dolphins.

For the ICI, the average and standard deviation are calculated only for the values that are below 200 ms, according to the distribution in the Fig. 3, thus eliminating the ICI that correspond to two distinct series of clicks.

Study	Our study	Rojas-Mena (2009)	Kyhn et al. (2010)	Marino et al. from
Reyes Reyes et al. (2018)
(unpublished data)	Martin et al. (2024)	
Number of clicks analyzed N	14,451	1,161	87	2,101	63	
Peak frequency	131 ± 11 kHz	126±2 kHz	126±3 kHz	154 ± 36 kHz	127 ± 4 kHz	
Centroid frequency	138 ± 9 kHz	126±2 kHz	129±3 kHz	146 ± 15 kHz	131 ± 5 kHz	
‘rms’ bandwidth	16 ± 5 kHz	10±3 kHz	12±3 kHz	Unavailable	11 ± 3 kHz	
−3 dB bandwidth	5 ± 2 kHz	15±3 kHz	15±4 kHz	9±4 kHz	10 ± 7 kHz	
−10 dB bandwidth	12 ± 6 kHz	26±7 kHz	Unavailable	21±11 kHz	Unavailable	
‘rms’ duration	79 ± 46 μs	Unavailable	Unavailable	Unavailable	Unavailable	
−10 dB duration	74 ± 42 μs	Unavailable	92±2 μs	57±33 μs	98 ± 22 μs	
−20 dB duration	160 ± 83 μs	124±36 μs	Unavailable	Unavailable	Unavailable	
Inter clicks interval (ICI)	69 ± 41 ms	31±10 ms	Unavailable	Unavailable	Unavailable	

The data presented in Rojas-Mena (2009), Kyhn et al. (2010) and Martin et al. (2024) are quite similar. Our study and the data presented in Reyes Reyes et al. (2018) present notable differences with these data (Table 2) and usually have much larger statistical dispersion. These differences may have several causes. There may be a intrinsic difference between the emissions of different groups of Peale’s dolphins, depending on environment and context. These differences have been shown in other NBHF click species as Harbour Porpoise (Phocoena phocoena) and Dall’s Porpoise (Phocoenoides dalli) (Kyhn et al., 2013).

Our study presents a larger number of clicks and thus, maybe, a greater diversity. This diversity influences the average of the clicks analyzed. To understand the distribution of the parameters, we draw the histograms of the parameters which have been presented in the preceding section, as in Patris et al. (2023) (see Fig. 3). The histograms of the peak frequency, of the ‘−10 dB’ duration, ‘−20 dB’ duration, and of the ‘−10dB’ bandwidth are multimodal so the characterization of these parameters using the average and the standard deviation (Table 2) is not really relevant. It should be remembered that this characterization (average/standard deviation) is very classic in many articles since Au (1993) but it is not relevant when the data follow a plurimodal statistical distribution.

Comparison of clicks parameters with acoustic studies of other NBHF species of Chile and Argentina

In Fig. 6 (left), we present the statistical repartition of peak frequencies for detected clicks of our data set. As in Patris et al. (2023), we fitted three Gaussian functions (oftypeS(f)=Ae−(f−fp)22σ2) to this histogram. The parameters (mean fp and standard deviation σ) of the three relevant Gaussian functions are presented in Table 3 along with the results given in Patris et al. (2023) and Reyes Reyes et al. (2015). We also checked for an influence of our subsets (visually confirmed and “on-axis” clicks) by fitting Gaussian functions to the histogram of the peak frequencies in these subsets. As can be seen in Table 3, the parameters of the Gaussian adjustment are very similar between the three selected series of values of our study (all the clicks, clicks visually confirmed, “on-axis” clicks). This confirms that there is no statistical difference between our subsets, and that our dataset is homogeneous.

Figure 6 Fitting of three or four Gaussian functions to the histogram of the peak frequency of the detections.

Left: Peale’s dolphins, data from our study, San Isidro, December 2021. Right: Probably Chilean dolphin, data from the Puyuhuapi Fjord, May 2021 (Patris et al., 2023). The two histograms do not have the same ordinate scale and contain a similar number of clicks.

Table 3 Gaussian fit ( fp±σ) to the histogram of the peak frequencies of the clicks in this study and in Patris et al. (2023) and median frequency ± interquartile range of the clusters given in Reyes Reyes et al. (2015).

Paper	This study all clicks	This study visually conf.	This study “on axis” clicks	Patris et al. (2023)	Reyes Reyes et al. (2015)	
Species	Peale’s dolphin (Lagenorhynchus australis)	Peale’s d.	Peale’s d.	Probably Chilean d. (Cephalorhynchus eutropia)	Commerson d.(Cephalorhynchus commersonii)	
Number of clicks	14,451	4,860	552	13,878	6,887	
Gausian fit no1	133 ± 2 kHz	132 ± 3 kHz	133 ± 2 kHz	136 ± 3 kHz	137 ± 6 kHz	
Gausian fit no2	122 ± 2 kHz	122 ± 2 kHz	122 ± 3 kHz	125 ± 4 kHz	129 ± 8 kHz	
Gausian fit no3	170 ± 10 kHz	169 ± 10 kHz	168 ± 9 kHz	168 ± 13 kHz	173 ± 14 kHz	

It should be noted that, after the Gaussian fit to the data, the standard deviation of the measures of peak frequency gets much closer to those of the data of Rojas-Mena (2009) and Kyhn et al. (2010) (see Tables 2 and 3).

The modes in Fig. 6 (left) are similar to those considered in Patris et al. (2023) (Fig. 6, right) where there is a good probability that the recorded clicks were clicks from Chilean dolphins (Cephalorhynchus eutropia). The modes also mostly coincide with the clusters presented in Reyes Reyes et al. (2015), where clicks from Commerson’s dolphins (Cephalorhynchus commersonii) were analyzed.

A large diversity of clicks for Peale’s dolphin is also found in Martin et al. (2024): NBHF clicks with a peak frequency around 128 kHz, broadband clicks with a peak frequency around 110 kHz and a small number of clicks with a peak frequency between 150 and 160 kHz. In Martin et al. (2024) article, only the first of these three manually separated categories has been analyzed for the quantitative results (as given in table 2). For consistency, we also discarded from the quantitative analysis the few trains of very high frequency clicks (150–160 kHz), which were manually detected. They were weak and could be due to poor conditions of sound propagation, very off-axis recording, or an intrinsically different kind of click.

In any case, all the studies tend to indicate the presence of three types of clicks with three typical peak frequencies. The fact that Peale’s and Commerson’s dolphins are not of the same genus indicates that two very different NBHF click species which are not in acoustic contact can have the same type of sound emission, in all its diversity.

In addition, the production of broad band and lower frequency clicks (“reducing the acoustic crypsis”) have been documented for some Cephalorhynchus NBHF clicks species as Heaviside’s dolphins (Cephalorhynchus heavisidii) or Commerson’s dolphins (Cephalorhynchus commersonii) (Martin et al., 2018, 2021) and recently have been proved for Peale’s dolphins (Martin et al., 2024) in the Falkland Islands. In our data set, no broadband and lower frequency clicks have been found, which could indicate that Peale’s dolphin do not relax acoustic crypsis in this zone, or at least not frequently. The Magellan Strait is a zone where killer whales (Orcinus orca) are present (Capella et al., 2014; Haro et al., 2023) and this could be a reason why this relaxing was not recorded in our data set. Another possible explanation is that, in most of the sightings, groups of Peale’s dolphins were slowly traveling, and NBHF classic high frequency clicks are probably more used in this context than lower frequency clicks that could be used in context of socialization.

Finally, in our data set, three atypical low frequency vocalizations were visually detected. These vocalizations come along with NBHF clicks and quick trains, and thus are associated with the presence of NBHF species. However, they are quite different from the whistles recently described by Martin et al. (2024). What is more, no whistle similar to those recorded by Martin et al. (2024) were detected in our experiment. This could be another clue indicating that the vocal repertoire of these animals may vary with place, ecological context, and/or the recording method.

Conclusion

The diversity of the repertoires recorded in Rojas-Mena (2009), Kyhn et al. (2010), Martin et al. (2024) and our study, with large variations between two places or distinct years in the same place tends to show that the repertoire of Peale’s dolphins is highly variable. It could result from the way the recordings have been made (Martin et al., 2024), the predator pressure, the season or the biotope (Kyhn et al., 2013). There is a strong need for long term recordings of this species, in distinct places, to better understand the influence of these parameters on the variations of their repertoire. We are also confronted to the difficulty of distinguishing acoustically between Peale’s dolphins and other NBHF species present in Patagonia: Chilean dolphin (Cephalorhynchus eutropia), Commerson dolphin (Cephalorhynchus commersonii) and Burmeister porpoise (Phocoena spinipinnis). New tools should be considered to classify NBHF clicks for each of these Patagonian species, in order to be able to assert the presence of each species during a PAM experiment.

Finally, this work shows the importance of conducting a preliminary medium-term study coupling visual monitoring, C-POD detectors and full wave high frequency hydrophones. It measures the efficiency of well-recognized C-POD in a specific context, and should be continued with long-term installation of such detectors to answer ecological interrogations, such as yearly presence, use of habitat and reaction to anthropological threats. The logistics are, however, rather difficult in such a remote area. We think that special efforts should be put by the international community in studying such poorly known species in territories with difficult access but rapidly evolving conditions such as the Magellanic strait.

Supplemental Information

Supplemental Information 1 Supplementary materials.

The figures and computations of this article were done in OCTAVE (Eaton, Bateman & Hauberg, 2009).

Additional Information and Declarations

Competing Interests

The authors declare that they have no competing interests.

Author Contributions

Franck Malige conceived and designed the experiments, performed the experiments, analyzed the data, prepared figures and/or tables, authored or reviewed drafts of the article, and approved the final draft.

Julie Patris conceived and designed the experiments, performed the experiments, analyzed the data, prepared figures and/or tables, authored or reviewed drafts of the article, and approved the final draft.

Benjamin Caceres conceived and designed the experiments, performed the experiments, analyzed the data, authored or reviewed drafts of the article, and approved the final draft.

Jonathan Poblete performed the experiments, analyzed the data, authored or reviewed drafts of the article, and approved the final draft.

Hervé Glotin conceived and designed the experiments, analyzed the data, authored or reviewed drafts of the article, and approved the final draft.

Monserrat Saravia performed the experiments, analyzed the data, authored or reviewed drafts of the article, and approved the final draft.

Constanza Alarcón-Vera analyzed the data, authored or reviewed drafts of the article, and approved the final draft.

Valentin Barchasz conceived and designed the experiments, authored or reviewed drafts of the article, and approved the final draft.

Valentin Gies conceived and designed the experiments, authored or reviewed drafts of the article, and approved the final draft.

Sebastian Marzetti conceived and designed the experiments, authored or reviewed drafts of the article, and approved the final draft.

Marjorie Fuentes-Riquelme conceived and designed the experiments, authored or reviewed drafts of the article, and approved the final draft.

Diego Filún conceived and designed the experiments, analyzed the data, authored or reviewed drafts of the article, and approved the final draft.

Data Availability

The following information was supplied regarding data availability:

The data is available at: https://sabiod.lis-lab.fr/pub/CHILI/SAN_ISIDRO_2021.

Patris, J. (2025). Soundscape and Peale’s dolphins. San Isidro, Chile, December 2021. [Data set]. Zenodo. https://doi.org/10.5281/zenodo.15078304.

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
