# Peer review of "Acoustic and visual monitoring of Peale’s dolphins (Lagenorhynchus australis) in the Magellan Strait"

_PeerJ, doi:10.7717/peerj.19234_

## Round 0.1 · original submission · Major Revisions

Dear Dr. Malige,

Although two reviewers have marked your paper as a minor revision, the first reviewer has identified some serious problems with the manuscript. We are therefore returning it to you for a major revision. Please pay particular attention to the following comments: !- all the results presented in the manuscript need to be discussed in the discussion part while now many parts are missing ; 2) one of the main problem of the clicks parameters is the sampling rate of the instruments; 3) missing of some methodologies in the section Material and methods.

Reviewer 1 ·

Basic reporting

The research article "Acoustic and visual monitoring of Peale’s Dolphins (Lagenorhynchus Australis) in the Magellan strait" is interesting as it describes the clicks parameters of a species not so studies. The english level is good and the number of references are sufficient. The introduction is well written while the discussion is too generic and doesn't give a deep interpretation of the results. The methodology and results are missing some parts (see my comments below).

Experimental design

The aim of the research is not very clear. There are many results presented but none of them is the real aim of the study. The authors are presenting the behavior of dolphins, number of clicks during day and night, comparison with other studies of the same species, comparison with other studies of different species, relationship between number of clicks and number of dolphins... I would keep things simple, focus on few of these things and decide the aim of the study. Moreover all the results presented in the manuscript need to be discussed in the discussion part while now many parts are missing.

I am also not convinced why the authors used 4 acoustic devices instead use only the HighBlue. In my opinion having different devices and different datasets make the results more confused. For example, the authors are presenting 3 datasets (total, visual confirmation and on-axis) very similar to each other, I would keep only one.

Validity of the findings

One of the main problem in my opinion of the clicks parameters is the sampling rate of the instruments. Your sampling rate is 512 kHz. Is it enough to catch the full click? When you compare your study with other studies, do you know their sampling rate? I would specify better this part in the manuscript.

Additional comments

These are my comments:
Abstract
Line 23: I would replace the word “Cone” with something else.
Line 25: replace “acoustic productions” with “acoustic behavior”.
Line 27: I would remove “several” and I would put the real number (4?)
Line 29: I would replace “acoustic productions” with “clicks parameters” or something like that. In the end you are comparing those parameters.

Introduction
The introduction is well written. I would just add some information on the dolphin acoustics. You can add some lines about clicks (what they are produced for…).
Line 59: In the manuscript you are saying that the species produces also whistles. Why do you use the term “NBHF”? Maybe you can remove the term and just say what sounds they produce (whistles and clicks mainly).
Line 62: What do you mean as “quality factor”. I would specify it or just remove it.
Lines 77: I would replace “vocal repertoire” as you are analyzing only the clicks and not all the repertoire.
Lines 78-79: I would remove the conservation sentence.

Material and methods
In this section some methodologies are missing (comparison with other studies, statistical analysis…). You presented in the discussion some results but everything should be described here first.
Line 81: I would add the hours of effort, “during four days (XXX hours)…”
Lines 91-92: I would add the ethogram in the supplementary material describing these behaviors
Line 92-93: if you didn’t use the drone, you can just remove it from the methods.
Line 97: Why did you decide to put 4 devices so close to each other? What is the sampling rate of the C-pod?
Line 136: As you said the sampling rate probably doesn’t consider the full click. So why did you decide to get these parameters? The peak frequency for example could change with different sampling rate

Results
Line 157: You can remove “(Lagenorhynchus australis)” as you already mentioned it.
Lines 159-160: What do you mean with “opportunistic sightings”? Citizen science?
Line 168: What are the % of all the behaviors? I would specify it.
Lines 168-170: notably…0h31 UTC. I would remove this sentence.
Line 170: (see figure 1, bottom right): in the caption the photo is from 10th of December while here it seems it is from 11th of December. I would remove the sentence because it is a bit confusing.
Line 184: No need to says the name of the file, I would remove that.
Line 190: “as in Patris et al. (2023)”. What do you mean?
I would remove Figure 3 as you are already giving the table 1 with the click parameters.
Lines 203-204: Did you perform visual monitoring during the full recording time of the HighBlue device? I mean 34% of the total clicks were classified as “with visual counterpart” but the remaining 66% were you doing visual monitoring and you didn’t see dolphins? Or some time you were not doing visual monitoring (like in the night)? I would specify it better in the text.
I would remove Figure 4 or move in the methodology while you describe the method and there you can show an example of clicks train.
Lines 228-238: This part of statistical analysis and models is missing in the methodology. From where did you get these 3 models? In my opinion this part should be removed from the study. You have very few groups of dolphins and from these results you cannot conclude the function of the clicks you have detected.
Why did you decide to have 3 datasets in Table 1? You are not statistically comparing the 3 datasets
Discussion
Lines 247-260: Here you are not commenting the particularity of your results. This part is too generic, I would discussed deeper your results more than generic observation (like having data during night hours or the limit of behavioral analysis from shore).
I would move Table 2 in the Results section. In the material and methods also is missing this comparison part. I would introduce it (what you compared and how) in the material and methods section too.
What is the sampling rate of the other studies? Different results could be due to different sampling rates.
Lines 266-268: You could use a statistical test (such as One-sample Wilcoxon test) to compare your results with the mean of the other studies. In this way you would have a more robust comparison.
Lines 269-270: Here I would add a reference of this species or other dolphin species that showed differences in clicks parameters between different groups.
Lines 281-285: This part should be in the Results section and the methodology explained in the Material and Methods section. Furthermore, I would use a statistical test to compare the studies more than the histograms. Why did you decide to compare the clicks of different species from different geographical areas?
Figure 6 and table 3 should go in the Results section. You should also explain what you did and how in the methodology.
Lines 305-315: I would remove this part. What do you mean with lower frequency clicks?
Lines 316-325: I would remove this part and I would focus on the clicks. If you don’t know what sound is better avoid conclusion and remove this part from the study.

Reviewer 2 ·

Basic reporting

This article provides valuable information to fill in research gaps for an otherwise poorly described species. The overall structure of the manuscript is well-organized. I have added some specific comments below.

Experimental design

No comment

Validity of the findings

No comment

Additional comments

Methods
Line 82: Just for consistency, UTC should be mentioned here too (11 h)
Figure 1: It would be valuable to provide the location of the recording units on the map, along with the depth contours of the area if available.
Line 120: When saying click trains usually come in a series of 10 clicks, would there be a reference for this? Is this specific to Peale's dolphins, or would you be saying this for species producing NBHF signals in general? Clarifying this would be useful for replicating the method.
Lines 120-123: Finally, to check... at least 20 detections. This sentence is a bit confusing; I would suggest breaking it up to simplify it.
Line 127: Were there any instances of recording only whistles from the dolphins in the absence of clicks?
Line 144: Data collection dates don't match, 8-12th or 8-11th?

Results
Line 195: How many such instances were recorded? Any potential reason why the detector did not pick these? It would be useful to add a bit on this in the discussion.
Line 222: It may be useful to mention the visual sighting conditions during the simultaneous events. If some correlation or statistical test was done between these detection types, that would be good to add in, too. I do understand that sample size may be an issue, but it would be interesting to see.
Line 226: Were any other sounds (whistles) from the species recorded during these 3 events?

Reviewer 3 ·

Basic reporting

The authors present novel information on the feasibility of long-term monitoring of Peale’s dolphins in the Magellan Strait using acoustic recordings. The paper is well-written and organized, with appropriate literature references. Below are suggestions to improve the manuscript:

Abstract
Line 29: The phrase “acoustic productions of this group” should align with the earlier reference to “population of Peale’s dolphins” to maintain consistency.

Introduction
Lines 35-37: The sentence on how increasing human activities make it challenging to study and monitor marine wildlife is unclear. The authors should clarify the link between human activities and the difficulty of monitoring wildlife.
Lines 53-55: Specify “acoustic presence” when discussing insights gained from this approach, as a species may be present but not vocally active.
After Lines 60-62: Provide more information on other narrow-band high-frequency (NBHF) species in the region and the importance of distinguishing between them. This additional context will emphasize the study's necessity.

Materials and Methods
Lines 81-82 and 86-87: Mention the UTC offset once (e.g., “18h local time (UTC – 3h)”) and avoid repeating the local and UTC times elsewhere.
Line 144: Reconcile the dates for sightings. If opportunistic sightings occurred, include them in the first paragraph of the Materials and Methods under visual and acoustic monitoring.

Results
Lines 159-161: Include opportunistic sightings (Dec 7–11) in the Materials and Methods section to clarify that “all sightings” refer to this period, not Dec 8–12.
Figure 2: Add the depth and total detection for each instrument near the title. Include a legend for the red areas to indicate moments of continuous effort. Clarify whether “number of clicks per hour” refers to the average.
Consider using black and white instead of blue and yellow for Figure 4.
Figure 5: Increase the thickness of the red lines representing the average and standard deviation.
Table 2: Correct the peak frequency for Martin et al. 2024 to 127 ± 4 kHz (not 126 ± 4 kHz).

Conclusion
Lines 332-335: Highlight the presence of other NBHF species in Patagonia in the Introduction to underscore the importance of distinguishing species signals for passive acoustic monitoring.

Experimental design

Materials and Methods
Line 95: Clarify the mooring setup. Is the water depth 10 meters, 10 meters from the seafloor, or 10 meters below the surface?
Lines 97-99: Is the “HighBlue” recorder positioned directly on the seafloor?
Figure 1 (Top Right): Explain if the buoy attached to the mooring (20 kg) is independent, part of the mooring, or a schematic of the line with the C-POD. Clarify whether the C-PODs are positively buoyant and do not require attachment to a line with a buoy or float.
Current Conditions: Provide details about currents and how they might affect the depth of instruments without a surface float.
Lines 108-109: Specify the technical reasons (e.g., memory storage, battery life) limiting recording durations.
Line 119: Define “file” (e.g., “a 9-minute WAV file”).

Validity of the findings

Results
Lines 199-201 and Figure 4: Add a normalized waveform and envelope of a single click from the pulse signal to confirm a burst-pulse call similar to Morgan et al. 2024. Change the ICI scale to milliseconds (ms) and, if possible, provide a longer signal matching examples from Morgan et al. 2024.

---

## Round 0.2 · accepted · Accept

Dear Dr. Malige,

I have assessed the revision myself, and I am happy with the current version. Thank you for the detailed responses to the comments of the reviewers and for the corrections made to the manuscript.
Now your manuscript is ready for publication.

Best regards,
Alexander

Note: The species name (australis) should be written in lower case in the title.